# Genome-Wide Identification of the PP2C Gene Family and Analyses with Their Expression Profiling in Response to Cold Stress in Wild Sugarcane

**DOI:** 10.3390/plants12132418

**Published:** 2023-06-22

**Authors:** Xing Huang, Yongsheng Liang, Ronghua Zhang, Baoqing Zhang, Xiupeng Song, Junxian Liu, Manman Lu, Zhenqiang Qin, Dewei Li, Song Li, Yangrui Li

**Affiliations:** 1Sugarcane Research Institute, Guangxi Academy of Agricultural Sciences/Sugarcane Research Center, Chinese Academy of Agicultural Sciences/Guangxi Key Laboratory of Sugarcane Genetic Improvement/Key Laboratory of Sugarcane Biotechnology and Genetic Improvement (Guangxi), Ministry of Agriculture, Nanning 530007, China; shmilyx023@163.com (X.H.); zhangronghua@gxaas.net (R.Z.); zbqsxau@126.com (B.Z.); xiupengsong@163.com (X.S.); liujunxian868@163.com (J.L.); mark0905@163.com (M.L.); qinzqcn@163.com (Z.Q.); ldw11023@163.com (D.L.); 2Nanning Institute of Agricultural Sciences, Nanning 530021, China; b9523@163.com

**Keywords:** wild sugarcane, type 2C protein phosphatase (PP2C), phylogenetic analysis, gene expression profiling, cold stress

## Abstract

Type 2C protein phosphatases (PP2Cs) represent a major group of protein phosphatases in plants, some of which have already been confirmed to play important roles in diverse plant processes. In this study, analyses of the phylogenetics, gene structure, protein domain, chromosome localization, and collinearity, as well as an identification of the expression profile, protein–protein interaction, and subcellular location, were carried out on the PP2C family in wild sugarcane (*Saccharum spontaneum*). The results showed that 145 PP2C proteins were classified into 13 clades. Phylogenetic analysis suggested that SsPP2Cs are evolutionarily closer to those of sorghum, and the number of SsPP2Cs is the highest. There were 124 pairs of *SsPP2C* genes expanding via segmental duplications. Half of the SsPP2C proteins were predicted to be localized in the chloroplast (73), with the next most common predicted localizations being in the cytoplasm (37) and nucleus (17). Analysis of the promoter revealed that *SsPP2C*s might be photosensitive, responsive to abiotic stresses, and hormone-stimulated. A total of 27 *SsPP2C*s showed cold-stress-induced expressions, and *SsPP2C27* (Sspon.01G0007840-2D) and *SsPP2C64* (Sspon.03G0002800-3D) were the potential hubs involved in ABA signal transduction. Our study presents a comprehensive analysis of the SsPP2C gene family, which can play a vital role in the further study of phosphatases in wild sugarcane. The results suggest that the PP2C family is evolutionarily conserved, and that it functions in various developmental processes in wild sugarcane.

## 1. Introduction

The phosphorylation and dephosphorylation of proteins are functionally reversible processes that can affect the growth and development of a variety of plants [1]. As important phosphatases, serine/threonine protein phosphatases (PSPs) were first classified into type 1 and type 2 in animals on the basis of their substrate specificity and pharmacological properties [2]. Type 2 phosphatase (PP2) functions to phosphorylate the alpha subunit of phosphatase kinase and is not regulated by inhibitors. PP2 can be further divided into three types, PP2A, 2B, and 2C, of which PP2C is regulated by Mg^2+^ [3]. The Mg^2+^-dependent PP2C protein is less abundant in mammals but ubiquitous in plants [4].

The PP2C protein is the main phosphatase family member in plants, accounting for 60–70% of the total phosphatase, such as 62.7% in *Arabidopsis thaliana* and 68.9% in *Oryza sativa* [5]. The high proportion of PP2C proteins suggests that they have important evolutionary significance in plants and may be involved in various cellular functions in plants. The PP2C phosphatase family in plants is considered a regulator of signal transduction and can be activated by a variety of environmental stresses or developmental signals to function [4,6]. More and more PP2C family proteins in plants have been identified and studied in recent years. There are 86, 94, 62, and 95 PP2C proteins in *Brachypodium distachyon* [7], *Medicago truncatula* [8], *Fragaria vesca* [9], and *Triticum aestivum* [10], respectively. Such a large number of PP2C proteins in plants indicates that they play an irreplaceable role in the growth and development of plants.

The PP2C family in rice and *Arabidopsis* has been further divided into 11 subfamilies (A–H) [5]. The PP2Cs share sequence conservation and similar evolutionary relationships in different plants [11]. A highly conserved motif located near the C-terminus of the catalytic domain of the PP2C protein of the D subfamily was found in *Arabidopsis*, which is an auxin-induced SMALL AUXIN UP RNA (SAUR) and is necessary for binding; hence, the mutation of this motif enables plants to express immunity to SAUR and SAUR–PP2C. Subfamily D can affect the proliferative process of plant cells by regulating PM-H1-ATPase activity [12]. *Arabidopsis* PP2C-type phosphatase 1 (AP2C1) interacts with certain members of the MAPK cascade (MPK3, MPK4, and MPK6), thereby regulating plant immunity and the response to hormones [13,14]. In *Arabidopsis*, ABI1 and ABI2 in subfamily A are the two most widely studied typical PP2C proteins. They play a regulatory role in the ABA signaling pathway and are important members in the ABA signal transduction process. They have integrated ABA and the function of Ca^2+^ signaling and phosphorylation-dependent response pathways. PP2Cs play a key role in ABA signal transduction and can modulate plant responses to different abiotic stresses by interacting with different target proteins [15]. The protein PP2C5 in *Arabidopsis* has been identified as a MAPK phosphatase, which colocalizes and interacts with stress-induced MPK3, MPK4, and MPK6 mainly in the nucleus, and changes in the expression level of PP2C5 can affect the expression of MAPK activation [16]. In addition, the PP2C38 protein in *Arabidopsis* can participate in the regulation of the phosphorylation of BIK1 protein and play a negative role in the immune response of plants [17].

A PP2C protein In alfalfa, MP2C (*Medicago sativa* phosphase 2C), has also been shown to be involved in the regulation of stress-induced MAPK cascades [18]. The overexpression of the maize ZmPP2C gene in *Arabidopsis* can reduce plant tolerance to salt and drought [19]; heterologous expression of the PP2C-a10 protein in wheat can regulate seed germination in *Arabidopsis* and adaptation to drought [20]. In addition, OsPP2C09 in rice positively regulates plant growth and negatively regulates plant drought resistance by participating in the ABA signaling pathway, and further studies have confirmed that, under drought stress, OsPP2C09-mediated ABA desensitization contributes to rice root elongation [21]; OsPP2C27 in rice can dephosphorylate OsMAPK3 and OsbHLH002, thereby negatively regulating plant tolerance to low temperatures [22]. The PP2C protein has various functions in plants, the most important of which is involved in regulating the tolerance of plants to stress.

Sugarcane is an important cash crop. Its cultivation can not only provide food and beverage needs, but also sugar, alcohol, and other industrial raw materials. Sugarcane cultivation plays an important role in promoting the development of local economy. Sugarcane is one of the world’s major industrial raw materials, with about 250 million hectares of sugarcane planted globally. Brazil, India, China, Thailand, and the United States are the world’s major sugarcane-producing countries. China is one of the major sugarcane producers in the world, and its sugarcane cultivation area is mainly distributed in the southern region. According to statistics, China’s sugarcane plantation area is about 600,000 hectares, mainly distributed in Guangxi, Yunnan, Hainan, Guangdong, and other provinces.

Although sugarcane has strong adaptability, its tolerance to low temperatures is not very strong, which directly causes certain limitations to the planting area of sugarcane. Under normal natural conditions, sugarcane production is limited by the weather. However, when sugarcane encountered rare, large-scale, continuous low-temperature, and freezing extreme weather, a survey in the contour zone at different levels found that the sugarcane growing point and the cane stem tissue were damaged or necrotic after the disaster.

The PP2C phosphatase family plays an important role in the process of the plant response to abiotic stress. On the basis of the sequenced genome data of sugarcane and combined with transcriptomic data analysis, this study aims to use bioinformatics methods to analyze the PP2C phosphatase family in sugarcane. At the same time, the mechanism of PP2C protein in sugarcane involved in the low-temperature stress response is explored to provide evidence for further understanding the mechanism of low-temperature tolerance in sugarcane and to provide a theoretical basis for improving varieties.

## 2. Results

### 2.1. Genome-Wide Identification of Sugarcane PP2C Gene

The PP2C domains of the candidate proteins we identified from the sugarcane genome were verified with the Pfam and SMART websites, and the results showed that there were 145 PP2C proteins in sugarcane; the basic information on these proteins, including the gene ID, protein size, and isoelectric point *p*I, is listed in Appendix A. A detailed analysis of the physicochemical properties of the SsPP2C family of proteins shows that the amino-acid sequence length of PP2C proteins in sugarcane ranges from 225 aa (Sspp2c140) to 1081 aa (Sspp2c73), and most SsPP2C proteins containing only one PP2C domain have an average length of 400 aa. In addition, the predicted molecular weight (MW) ranges from 24.30 kDa (Sspp2c140) to 119.98 kDa (Sspp2c73), and the isoelectric point (*p*I) varies from 4.58 (Sspp2c61) to 9.38 (Sspp2c32).

### 2.2. Phylogenetic Analysis, Gene Structure, and Protein Motif Analysis

In order to comprehensively understand the evolutionary relationship of PP2C family proteins in sugarcane, we used MEGA (version 7.0.26) to align the PP2C proteins of sugarcane, *Amborella trichopoda*, *Arabidopsis*, rice (*Oryza sativa*), foxtail millet (*Setaria italica*) maize (*Zea mays*), and sorghum (*Sorghum bicolor*) and compared them using MEGA. A phylogenetic tree was constructed on the basis of the alignment results (Figure 1).

According to phylogenetic analysis with *Arabidopsis* of PP2C family proteins, SsPP2C proteins were divided into 13 subfamilies. Among all the subfamilies, the subfamily with the largest number of SsPP2C proteins was the F subfamily, with 37 members; the second was subfamily D, with 30 SsPP2C proteins; there were 24 PP2Cs in subfamily A and 37 members in subfamily G; in the remaining subfamilies, there were 0–9 SsPP2Cs, and one SsPP2C protein was not classified into a subfamily (Figure 1 and Figure 2).

In addition, by comparing the distribution of PP2C proteins in different subfamilies in different species, it can be found that the distribution of PP2C proteins in different subfamilies was similar, and the proportion of PP2C proteins in different subfamilies was also similar, indicating that their functions were similar.

In the process of phylogenetic tree construction of sugarcane PP2C proteins and PP2C proteins of rice, maize, and sorghum, it was found that SsPP2C proteins tend to aggregate with SbPP2C proteins, while PP2C proteins of other species tend to form separate branches. Therefore, the PP2C protein is highly conserved in different species, and the PP2C protein in sugarcane is more closely related in evolution to that in sorghum (Appendix A).

To investigate whether the structures of sugarcane *PP2C* genes are similar in different subfamilies, we analyzed the CDS sequence and genome sequence of the *SsPP2C* gene by comparing the structure of the *SsPP2C* gene (Figure 2). The results showed that the number of exons of the sugarcane *PP2C* gene varied from one to 14, with an average of six. In addition, *SsPP2C* genes belonging to the same subfamily tended to have the same exon/intron structure pattern, and almost every member of the subfamily had a high degree of similarity in exon number and even exon length. This shows that these proteins may have similarities in their functions.

To further understand the SsPP2C protein, we used the online software MEME to detect these conserved protein motifs; the results identified a total of 10 conserved motifs (Figure 2 and Figure 3), and it was found that the SsPP2C proteins belonging to the same subfamily have similar motifs and sequence modes. Furthermore, motifs 2, 3, 4, and 5 formed the basic PP2C domain, whereas motif 8 occurred only in subfamily D proteins. Lastly, the analysis of the gene structure and protein motifs also showed that the composition of theSsPP2C motif was consistent with its gene structure.

### 2.3. Analysis of cis-Elements in the Promoter Region of Sugarcane PP2C Gene

The function of the gene is directly linked to the *cis*-acting elements in the promoter region; thus, in order to study the potential function of the *SsPP2C* genes, we also analyzed the promoter regions of these *PP2C* genes. The *cis*-acting elements contained in the 2 kb genome sequence upstream of the *SsPP2C* gene transcription initiation site was analyzed by searching PlantCARE online database (http://bioinformatics.psb.ugent.be/wedescriptionbtools/plantcare/html/, accessed on 6 May 2021) (Appendix A), and 11 of the 122 *cis*-elements are shown in Appendix A.

In this study, we detected a total of 122 functionally relevant *cis*-acting elements including common eukaryotic regulatory elements such as TATA-box, and the details of these 122 *cis*-acting elements are shown in Appendix A. Statistical analysis of the element categories contained in the promoter regions of all *SsPP2C* genes found that, in addition to a large number of light-responsive elements in the promoter regions of all *SsPP2C* genes, more than half of the promoter regions of sugarcane *PP2C* genes contained different plant hormones. Response elements, such as the MeJA, ABA, and SA response elements, indicated that most of the PP2C proteins in sugarcane may be involved in the regulation of hormone responses. In terms of abiotic stress, we also found that almost all *SsPP2C* gene promoter regions have light-inducible response elements, and more than half of the *SsPP2C* gene promoter regions contain drought-inducible response elements and low-temperature response elements, indicating that the SsPP2C protein family is involved in the process of sugarcane defense against a variety of abiotic stresses. In addition, it was observed that very few *SsPP2C* genes may be involved in the regulation of flavonoid biosynthesis genes and the regulation of plant damage.

### 2.4. Chromosomal Location, Gene Duplication, and Collinearity Analysis

To analyze the chromosomal distribution of *SsPP2C* genes, we mapped each gene and marked its location on the corresponding chromosome (Figure 4). The results showed that 145 *SsPP2C* genes could be located on 32 chromosomes of sugarcane. The *SsPP2C* genes located on chromosomes were unevenly distributed on chromosomes, and Chr1A was the most abundant (11), followed by Chr1B (9), Chr5C (8), Chr1D (7), etc. From the haplotype genome, the numbers distributed on the A, B, C, and D genomes were 39, 38, 37, and 31, respectively. There was only one *SsPP2C* gene in Chr6 of each sub-genome. These results suggest that the *SsPP2C* gene is not only unevenly distributed within each subfamily but also unevenly distributed across different chromosomes.

In addition, after rigorous screening, we found a total of 124 pairs of symbiotic homologous genes in the sugarcane *SsPP2C* family, involving 108 *SsPP2C* genes; they were all found to be segmental duplication events, and no tandem duplication gene pairs were found (Figure 5). More importantly, this indicated that the increase in the number of *SsPP2C* gene families was mainly due to chromosome polyploidy, especially concentrated on chromosomes 3, 4, and 5, followed by chromosomes 1 and 8.

In order to explore the homology of the *PP2C* gene in sugarcane and other plants, we conducted a homology analysis on the *PP2C* gene in sugarcane and another three species. The results showed that a total of 130 *SsPP2Cs* genes have orthologs in rice (81), sorghum (107), and maize (104). The results showed that the SsPP2C gene showed high collinearity with the three species, and the SsPP2C gene had the highest homology with the PP2C gene of sorghum, followed by maize (Figure 6).

### 2.5. Prediction of Subcellular Localization of SsPP2C Protein

Subcellular localization prediction indicated that the SsPP2C protein was mainly distributed in the chloroplast (74), cytoplasm (37), and nucleus (17), followed by the extracellular matrix mitochondria and peroxisomes (Appendix A). Nearly 50% of PP2C proteins were predicted to be localized on the chloroplast, which indicated that the SsPP2C protein may play a role in photosynthesis signal transduction. Furthermore, SsPP2C proteins belonging to one subfamily tend to localize to the same subcellular location. The above results not only proved the high similarity of the proteins of each subfamily but also provided evidence for the further study of the function of the SsPP2C protein.

### 2.6. Functional Analysis of SsPP2C Genes under Cold Conditions

According to the recently sequenced transcriptome data of the gene expression level in wild sugarcane after low-temperature treatment, a total of 27 *SsPP2Cs* were screened out at the transcriptional level in response to cold tolerance (Appendix A; Figure 7). Compared with the control, eight *SsPP2C* genes were upregulated and 18 were downregulated during the cold treatment for 0.1, 1, and 6 h. After exposure to low-temperature treatment for 0.5 h and 1 h, 24 *SsPP2C* genes had a remarkable tendency to recover to the expression level of untreated specimens (control). *SsPP2C1* showed continuous upregulation under cold conditions from 1 h to 6 h, while *SsPP2C91* and *SsPP2C97* were downregulated until 6 h of cold conditions.

We selected 10 genes for qRT-PCR and conducted linear correlation analysis. The results showed that there was a high correlation R^2^ = 0.9229 between the two methods (Appendix A). Furthermore, we performed qRT-PCR in the time course of expression of PP2C genes during cold stress exposure, which further confirmed the high credibility of transcriptome data (Appendix A).

### 2.7. Construction of Protein–Protein Interaction Network of SsPP2C Protein

We predicted the network diagram of their interaction among all the SsPP2C proteins using STRING (Figure 8). The results showed that these low-temperature-induced SsPP2C proteins were related to calcium signaling, MAPK cascade, ABA, light, and other processes in sugarcane cells. In addition, SsPP2C74 (Sspon.04G0026520-1T) was a potential hub protein among these SsPP2Cs, with 42 possible linear interacting proteins; it may simultaneously participate in calcium, the ABA signaling pathway, MAPK cascade, and the light-mediated signal transduction pathway.

Gene expression profiles generally reflect the biological function. According to the results of transcriptome sequencing of sugarcane genes after low-temperature (4 °C) treatment in our laboratory, we screened a total of 27 *SsPP2Cs* that responded to low-temperature stress, which exhibited a differential expression pattern (Figure 7). In addition, we predicted the network diagram of their interaction using STRING (Figure 9). The results showed that five low-temperature-induced SsPP2C proteins (*SsPP2C27*, *SsPP2C64*, *SsPP2C31*, *SsPP2C110*, and *SsPP2C144*) had a strong interactional relationship and *SsPP2C27* (Sspon.01G0007840-2D) and *SsPP2C64* (Sspon.03G0002800-3D) were potential hub proteins among these *SsPP2Cs*, involved in ABA signal transduction through interaction with PYL and DIP proteins.

### 2.8. Subcellular Localizations of SsPP2C Proteins

To confirm the subcellular locations of SsPP2C proteins predicted by Plant-mPLoc, SsPP2C27, SsPP2C31, SsPP2C64, and SsPP2C110 were selected from three classes to perform subcellular localization analysis. As shown in Figure 10, the green fluorescent protein (GFP) signals of the positive controls were observed throughout the protoplasts, and the same distribution feature of fluorescent signals for SsPP2C27, SsPP2C64, and SsPP2C110 was observed. On the other hand, fluorescent signals of SsPP2C31 were found at the margin of the protoplast, and the fluorescence signal intensity was significantly higher than that of other fusion proteins, suggesting that the SsPP2C31 protein might be localized in the cytoplasm and plasma membrane.

## 3. Discussion

The PP2C phosphatase family is the most abundant phosphatase family in plants. It is evolutionarily conserved and participates in the regulation of various plant growth and development processes, especially in plant adaptation to stress. PP2C-type phosphatases are considered to be a representative family of PPM phosphatases. PP2Cs are widely distributed in eukaryotes. A total of 16 different PP2C genes were identified in the human genome, encoding at least 22 isozymes, mainly involved in cell growth and cellular stress signaling, and PP2Cs can act as cells in almost all stress-related situations and as inhibitors of stress signaling. Researchers have identified a total of 76 PP2C proteins in *Arabidopsis* [23], accounting for more than 60% of *Arabidopsis* protein phosphatases.

This study provides a comprehensive analysis of the PP2C family in sugarcane based on the sequenced sugarcane genome data. The results show that the PP2C proteins in sugarcane were divided into 13 subfamilies with a total of 145 members, which were similar to those in maize and sorghum and much higher than those in *Amborella trichopoda*, *Arabidopsis*, millet, and rice. These results indicate that both genes underwent a wide expansion in sugarcane compared to *Arabidopsis*. Furthermore, in lower plants such as *Chlamydomonas reinhardtii* and mosses such as *Physocmitrela patens*, the genome size is comparable to higher plants such as *Arabidopsis* and rice [24,25,26,27], but the quantity of PP2C proteins varies widely, such as 10 in green algae, 50 in bryophytes, 70–130 in plants such as *Arabidopsis*, rice, and maize [23], and only 48 in *Amborella trichopoda*. It can be seen that the expansion and diversification of the *PP2C* gene family may be closely related to the evolutionary process of plants from lower to higher.

SsPP2C members in the same group exhibit relatively the similar molecular weight and *p*I values, while different groups possess different pIs and molecular weight. These variations would lead to changes in gene structure and constitution of the protein domain, playing important roles in the PP2C family’s evolution.

The results of gene (intron–exon) structure analysis of the *PP2C* family of different plants showed that the *PP2C* genes in different plants have a certain degree of conservation in structure. For example, the number of introns in *Arabidopsis* ranges from zero to 12, while the rice PP2C gene contains 0–18 introns [5,28], that of upland cotton (*Gossypium hirsutum*) contains 1–10 introns [29], and that of turnip (*Brassica rapa*) contains 0–19 introns [30]. In addition, members belonging to the same subfamily have certain similarities in their intron–exon structure and gene length, indicating their close evolutionary relationship and conservation of the gene structure among different plant species. On the other hand, researchers have detected the occurrence of many chromosomal duplication events in both *Arabidopsis* and rice *PP2C* gene families, indicating that chromosomal duplication has a role in promoting the expansion and evolution of *PP2C* genes in the process of plant evolution [5]. In addition, the chromosomal duplication event in rice can not only promote the expansion of the *PP2C* gene but also plays a certain role in the diversification of the *OsPP2C* gene family function, because, in addition to some duplicated gene pairs showing conservation, other duplication pairs exhibit neofunctionalization and pseudo functionalization [28]. From the results of this study, polyploidy is the main reason for the sharp increase in the number of PP2C genes in sugarcane.

The function of a protein is often closely related to the domains it contains. Therefore, in order to understand the function of PP2C protein, it is of great significance to analyze its domains. Domain analysis of PP2C proteins in eukaryotes revealed that the PP2C domain can appear at both the N- and the C-termini [11]. For example, most of the PP2C proteins in *Arabidopsis* have the catalytic domain at the C-terminus, while a few *Arabidopsis* PP2Cs, especially those belonging to the F group, have the catalytic domain at the N-terminus [4]. These structural differences may indicate that their functions are also diverse. Detailed sequence analysis of the domains of PP2C proteins can not only reveal some unique domains and motifs in plant PP2C proteins but also allow a preliminary analysis of their possible functions. For example, the kinase-interacting motif (KIM) domain, a motif that can interact with MAPK, is present in most members of the *Arabidopsis* B family PP2C and regulates plant immunity and the response to hormones [13,14]. Furthermore, KIM motifs are found in the PTP proteins of most organisms and are essential for the functional activity of PP2Cs in regulating MAPK signaling in plants. In addition to the KIM motif, another conserved domain in PP2C protein is the fork point-associated domain (FHA). The FHA domain, the only known phosphorylated protein-binding domain, is present in a variety of proteins with diverse functions and is especially prevalent in proteins involved in DNA damage responses [31,32].

At present, studies have confirmed that many PP2C protein family members can participate in the regulation of various plant adaptation mechanisms to stress [20,21,22,33,34,35,36]. For example, in rice, OsPP2C27 can dephosphorylate OsMAPK3 and OsbHLH0002, thereby negatively regulating the adaptation of rice to low temperatures [22]. In *Arabidopsis*, the PP2C-type phosphatase PIA1 (PP2C induced by AvrRpm1) has been shown to be involved in the regulation of defense responses [33]. The PP2C-a10 protein in wheat can play a role in seed development and plant drought tolerance after heterologous expression in *Arabidopsis* [20]. PP2Cs play a key role in ABA signal transduction and can modulate plant responses to different abiotic stresses by interacting with different target proteins. In *Arabidopsis*, the core model of the ABA signaling pathway, the PYL(PYR/PYL/RCAR)–PP2C–SnRK2 model, was also identified, which catalyzes the phosphorylation of the corresponding downstream effectors, thereby activating the relevant stress in plants [34]. MAPK cascades can play an important role in the response to cold stress in plants [35]. ABA-dependent cold-responsive gene expression is also considered to be an important pathway for plants to respond to cold stress [36]. Therefore, whether PP2C protein, as a regulatory protein related to two signaling pathways, plays a role in the low-temperature signal response, as well as its mechanism of action in the low-temperature signal response, can be further studied and has important theoretical significance. In addition, OsPP2C09 in rice negatively regulates plant drought resistance [21]. OsPP2C27 in rice negatively regulates plant tolerance to low temperatures [22]. In our study, 32 SsPP2Cs responded to cold stress, and five of them were predicted to have an interaction with PYL and DIP proteins, which verified their potentially important roles in the core model of ABA signaling pathway PYL(PYR/PYL/RCAR)–PP2C–SNRK2 [34]. It can be seen that PP2C protein has various functions in plants, the most important of which is its involvement in regulating the tolerance of plants to stress. However, the role of PP2C protein involved in sugarcane cold stress needs further clarification.

## 4. Materials and Methods

### 4.1. Identification, Classification, and Protein Characteristics of PP2C Protein in Wild Sugarcane

To identify members of the PP2C phosphatase family in sugarcane, we searched the publicly published sugarcane genome database with the keyword “PP2C”, followed by using the Pfam (http://pfam.xfam.org/, accessed on 7 May 2021) and SMART (http://smart.embl-heidelberg.de/, accessed on 7 May 2021) websites to test whether the candidate proteins contained the PP2C domain. The amino-acid sequences and PP2C proteins of *Amborella trichopoda*, *Arabidopsis*, *O. sativa*, *Sorghum bicolor, Zea mays*, and *Setaria italica* were downloaded from the EKPD website (http://ekpd.biocuckoo.org/, accessed on 7 May 2021). The CDS sequence and the PP2C domain were verified in the same way as the sugarcane PP2C protein. The compute *p*I/MW tool of the ExPASy Server [37] was used to calculate protein properties such as the molecular weight (MW) and theoretical isoelectric point (*p*I). The ProtComp website (http://www.softberry.com/, accessed on 7 May 2021) was used to analyze the subcellular localization of the SsPP2C protein. In addition, the PP2C protein sequences of the above plants were subjected to evolutionary analysis [38].

### 4.2. SsPP2Cs Phylogenetic Tree, Gene Structure, Conserved Motifs, and Protein Domain Analysis

Homology analysis of SsPP2Cs was carried out as described in previous studies [39]. The SsPP2C, ArPP2C, AtPP2C, OsPP2C, ZmPP2C, SiPPP2C, and SSsPP2C protein sequences were aligned using the Clustal W program [40] and phylogenetically analyzed using MEGA X [41], followed by the JTT amino-acid substitution model [42], which uses the maximum likelihood (ML) to construct an unrooted phylogenetic tree. The evolutionary tree was beautified and visualized by the online software Evolview (V3) [43,44]. The online gene structure display server (GSDS: http://gsds.cbi.pku.edu.ch, accessed on 8 May 2021) was used to study the gene structure of *SsPP2C* gene family members [45]. The MEME (version 5.0.4) online search tool [46] was used to identify conserved motifs among related proteins in the *SsPP2C* gene family, and the number of selected motifs was defined as 10 except for the default setting. In addition, we utilized the InterPro database (http://www.ebi.ac.uk/interpro/, accessed on 8 May 2021) to predict the conserved domains of the SsPP2C protein family [47] and visualized them using Evolview online software (V3).

### 4.3. Analysis of Cis-Acting Elements of SsPP2Cs Promoter

To identify cis-acting elements in the promoter sequences of sugarcane *PP2C* family genes, the genomic sequence 2000 bp upstream of the transcription start site was analyzed using the online PlantCARE database [48]. Then, the obtained *cis*-acting elements were classified and counted, and the statistical results were drawn as a histogram using sigma plot 12.5 software. In addition, Venn diagrams between different kinds of *cis*-acting elements were drawn using the online Venn diagram drawing website Draw Venn Diagram (http://bioinformatics.psb.ugent.be/webtools/Venn/, accessed on 9 May 2021).

### 4.4. Chromosome Localization and Collinearity Analysis

We used TBtools (V1.098) to extract the location of the *SsPP2C* gene from the genome annotation gff3 file and then used MCScan X [49] to carry out collinearity analysis of members of the sugarcane PP2C family, as well as three other plants (rice, maize, and sorghum). Chromosome localization images, collinearity analysis images, and synteny analysis images were obtained.

### 4.5. Transcriptional Analysis of SsPP2C Genes under Cold Stress of Wild Sugarcane

The transcriptome data used in this paper were all obtained by screening the existing sequencing data in our laboratory (http://www.ncbi.nlm.nih.gov/sra/, accessed on 10 May 2021, accession number PRJNA636260). The method involves screening the gene numbers of sugarcane *PP2C* genes from all gene differential expression databases, extracting their expression, and using the FPKM value for normalization.

### 4.6. Protein–Protein Interaction Analysis of Transcriptome Data

The interaction among all sugarcane PP2C proteins and the differentially expressed *SsPP2Cs* after low-temperature treatment was predicted online using STRING (http://string-db.org/, accessed on 10 May 2021), a website for searching interacting proteins. Confidence was set to a high level (>0.4). The protein–protein interaction network was then plotted and beautified using Cytoscape (version 3.8.0) (http://www.cytoscape.org/, accessed on 10 May 2021).

### 4.7. Determination of Subcellular Localization of SsPP2C Proteins

The subcellular localization sites were predicted by Plant-mPLoc (http://www.csbio.sjtu.edu.cn/bioinf/plant-multi/, accessed on 10 May 2021). The CDS sequences of SsPP2C27, SsPP2C31, SsPP2C64, and SsPP2C110 without a termination codon were fused to the N-terminal of GFP under the control of the CaMV 35S promoter in the pBI121 vector using an In-Fusion^®^ HD Cloning Kit (Takara, Dalian, China). The fusion constructs were delivered into rice protoplasts. After incubation for 12–18 h, The GFP fluorescent images were photographed with a confocal laser scanning microscope (OLYMPUS, Tokyo, Japan, Olympus FV3000 viewer).

### 4.8. qRT-PCR Analysis

Total RNA was extracted from the control and cold-treated plants after 6 h using Trizol reagent (Invitrogen, Carsbad, CA, USA) and then digested with DNase I (TaKaRa) at 37 °C for 30 min to remove contaminating DNA. The qRT-PCR reaction was carried out in an Mx3000PTM Real-Time PCR system (Stratagene, La Jolla, CA, USA). qRT-PCR was performed in a 20 µL reaction mixture containing 2 µL of template cDNA, 10 µL of SYBR Green Realtime PCR Master Mix (TaKaRa, Dalian, China), 0.4 µL ROX as reference dye, 0.4 µL (10μM) of each primer, and 6.8 µL of RNA-free water.

The PCR reaction was run as follows: 94 °C for 5 min, and then 40 cycles of 5 s at 94 °C, 30 s at 60 °C, and 20 s at 72 °C, followed by single cycles of 95 °C for 1 min, 55 °C for 30 s, and 95 °C for 30 s for the dissociation stage. The expression levels of the transcripts were normalized to the endogenous genes *GAPDH*. Each set of experiments was replicated three times. The primers were designed for our selected genes with Primer3 primer (www.bioinfo.ut.ee/primer3-0.4.0/, accessed on 20 May 2023); refer to Appendix A for primer sequences. Relative expression levels of target genes were calculated using the 2^−ΔΔCt^ method. In order to further confirm the credibility of transcriptome data, an Excel table was used to analyze the linear correlation between qRT-PCR results and transcriptome results of selected genes.

## 5. Conclusions

In conclusion, a total of 145 PP2C family proteins in wild sugarcane were identified and classified into 13 clades. Phylogenetic analyses revealed that SsPP2Cs were evolutionarily closer to those of sorghum, and the number of SsPP2Cs was the highest. A total of 124 pairs of *SsPP2C* genes were found, expanding via segmental duplications. Half of the SsPP2C proteins functioned in the chloroplast (73), with the next largest localization being in the cytoplasm (37) and nucleus (17). The *cis*-acting element analysis of the promoter region of *SsPP2C* genes revealed that SsPP2Cs might be photosensitive, responsive to abiotic stresses, and hormone-stimulated. A total of five low-temperature-induced SsPP2C proteins had a strong interactional relationship, and SsPP2C27 and SsPP2C64 were potential hub proteins among these SsPP2Cs, involved in ABA signal transduction under cold stress. Our study presents a comprehensive analysis of the *SsPP2Cs* gene family; hopefully, it can help us to better understand the mechanism of sugarcane adaptation to the environment and lay the foundation for improving varieties.

Taken together, our findings suggest that the candidate SsPP2Cs should be the critical regulator of cold stress tolerance, which provide important insights into the molecular mechanism underlying cold stress response in sugarcane. However, there are few reports about the transgenic studies and functional identification of *PP2C* genes regulating cold stress tolerance, which may be due to the lack of an efficient genetic transformation system. For future study, transgenic studies should be performed on the candidate genes identified in present study, to further verify the underlying molecular mechanisms of these genes regulating cold stress tolerance.

## Figures and Tables

**Figure 1 plants-12-02418-f001:**
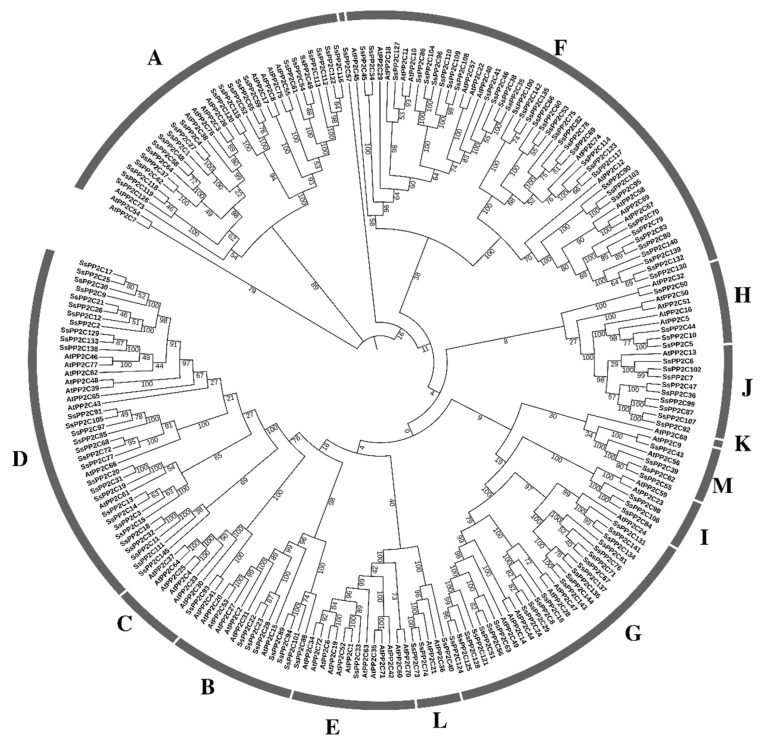
Phylogenetic analysis of PP2C proteins among wild sugarcane and *Arabidopsis*.

**Figure 2 plants-12-02418-f002:**
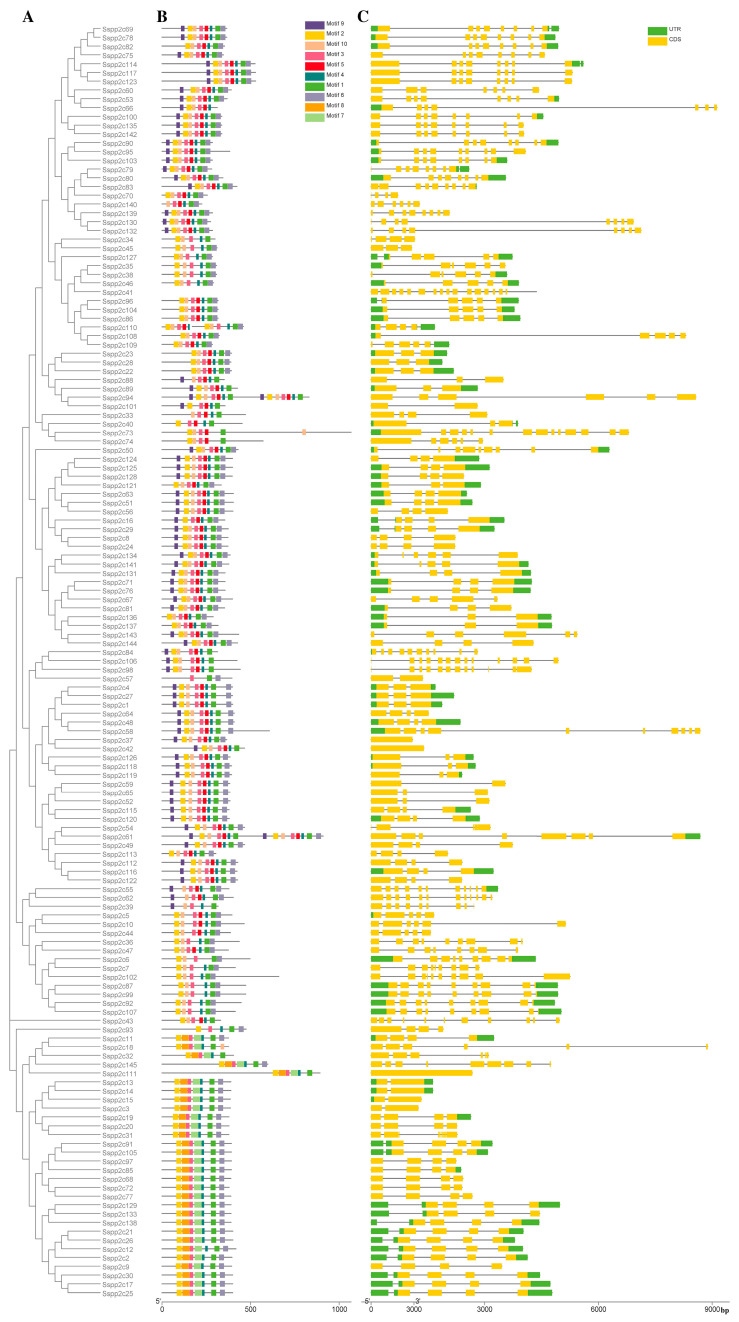
Analysis of gene structure and protein conserved motif of Sspp2Cs: (**A**) phylogenetic tree of **Ss**PP2Cs; (**B**) gene structure; (**C**) conserved motifs.

**Figure 3 plants-12-02418-f003:**
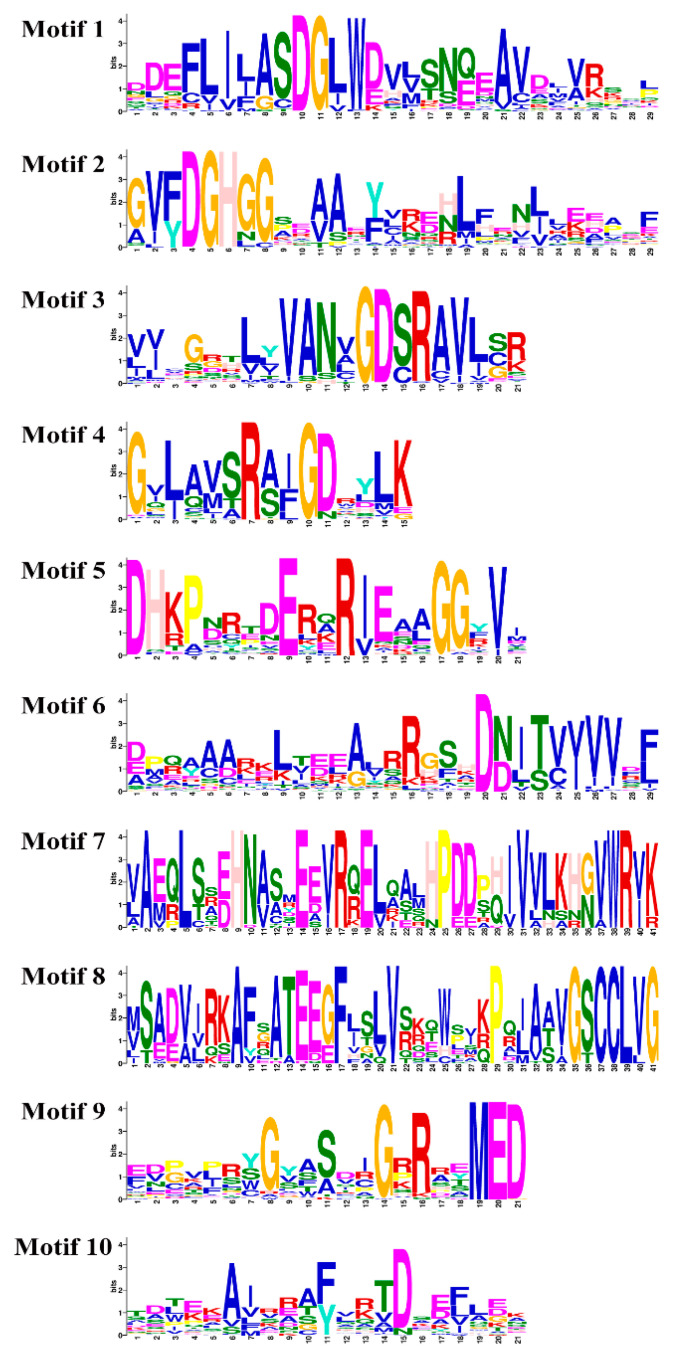
Conserved motif sequence of SsPP2C proteins.

**Figure 4 plants-12-02418-f004:**
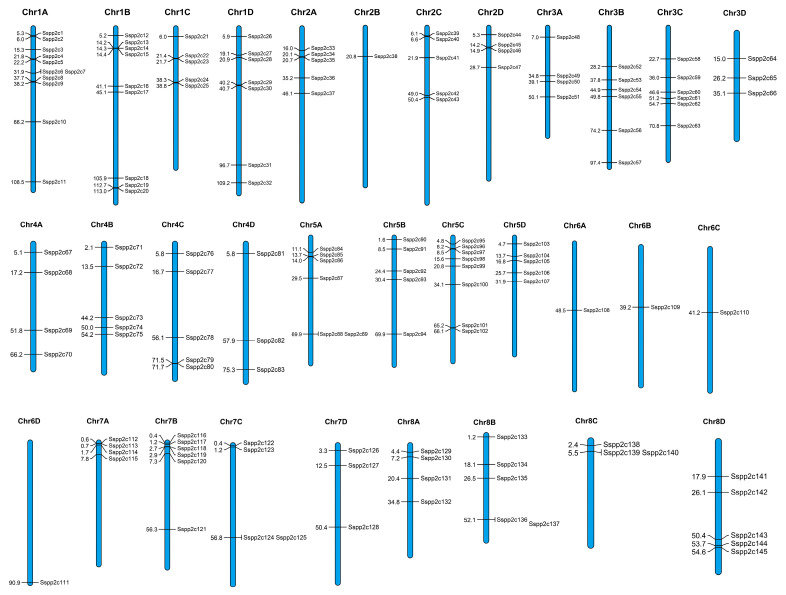
Chromosomal locations of *SsPP2C* genes.

**Figure 5 plants-12-02418-f005:**
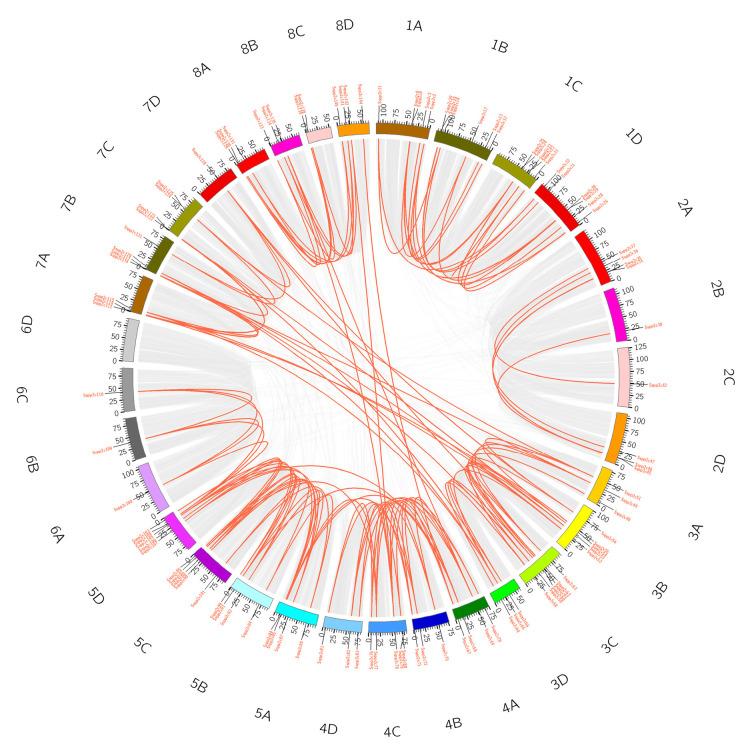
Duplication analysis of *SsPP2C* genes.

**Figure 6 plants-12-02418-f006:**
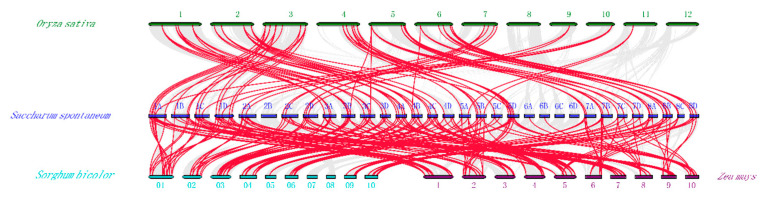
Synteny analysis among *PP2C* genes in wild surgacane and other plant species.

**Figure 7 plants-12-02418-f007:**
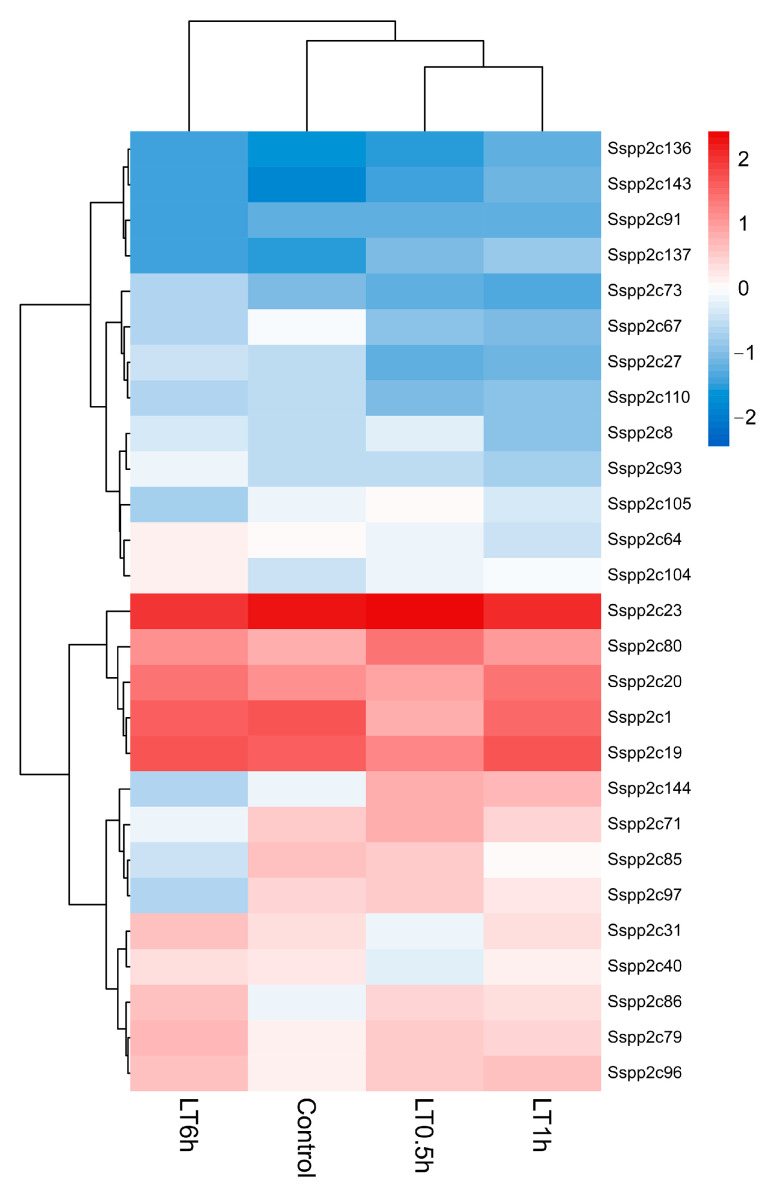
The heatmap of *SsPP2C* genes responding to low temperatures at transcription.

**Figure 8 plants-12-02418-f008:**
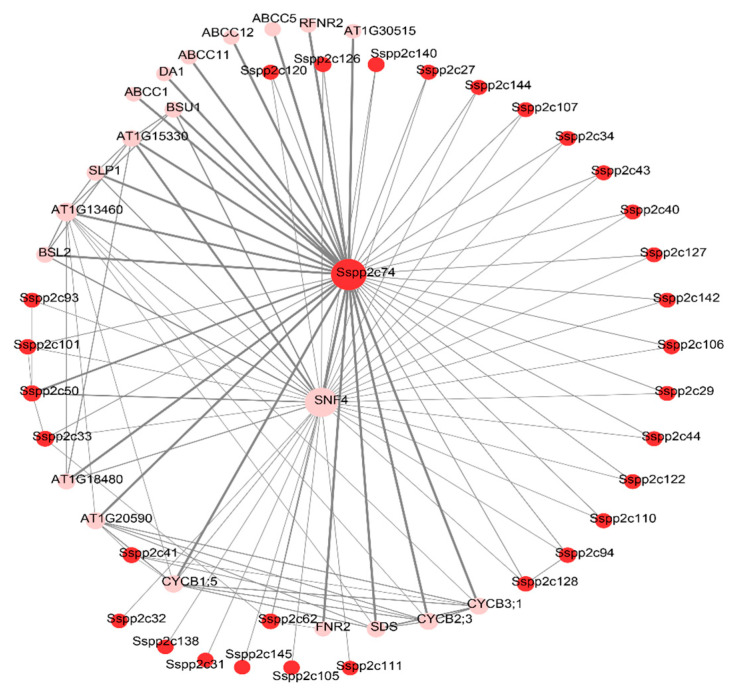
Protein–protein interaction analysis of all *SsPP2C*s.

**Figure 9 plants-12-02418-f009:**
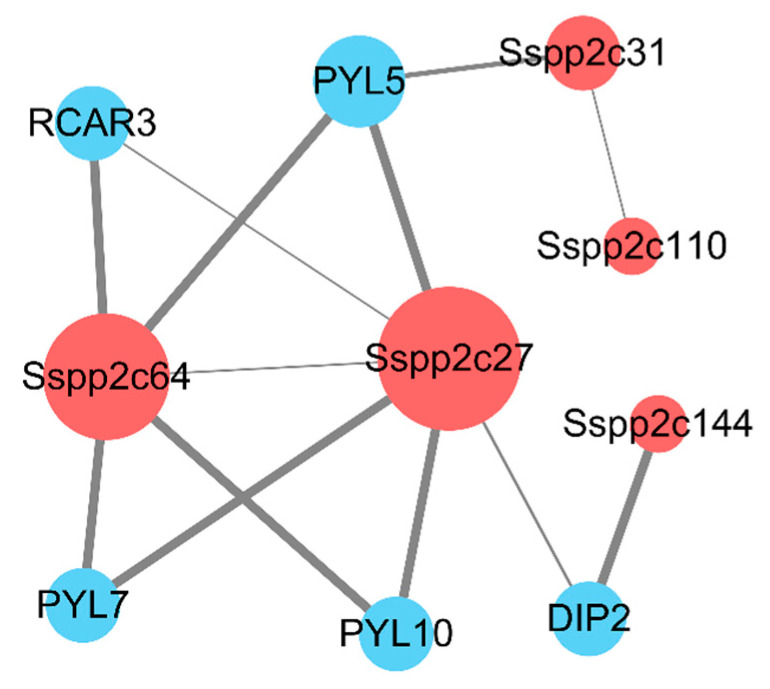
The protein–protein interaction analysis of *SsPP2Cs* responding to low temperature.

**Figure 10 plants-12-02418-f010:**
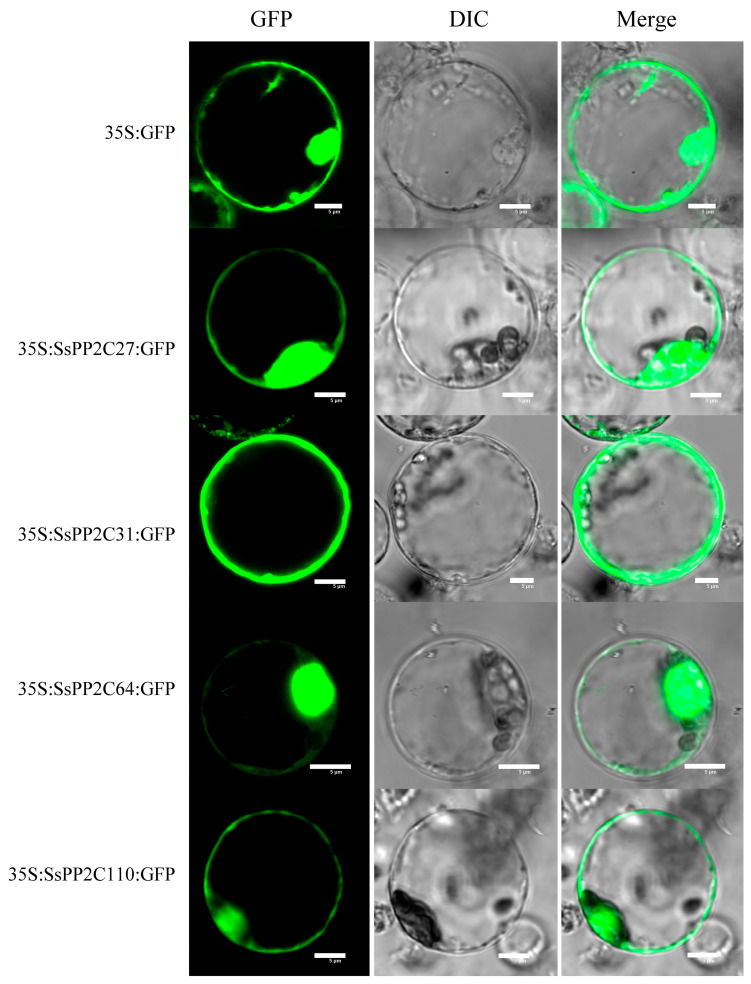
Subcellular localization of SsPP2C27, SsPP2C31, SsPP2C64, and SsPP2C110 in rice leaf protoplasts expressing green fluorescent protein (GFP). Scale bars represent 5 μm.

## Data Availability

The transcriptome data used in this paper were all obtained by screening the existing sequencing data in our laboratory (http://www.ncbi.nlm.nih.gov/sra/, accession number PRJNA636260).

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
