# Peer review of "Genome-Wide Identification of the PP2C Gene Family and Analyses with Their Expression Profiling in Response to Cold Stress in Wild Sugarcane"

_plants, 2023, doi:10.3390/plants12132418_

Round 1

Reviewer 1 Report

This article revealed Genome-wide identification of the PP2C gene family and analyses with their expression profiling in response to cold stress in wild sugarcane. Before recommending this article for publication, there are some shortcomings for that should be resolve.

Abstract is well written however information of specific methods are not well presented.

Line 20 “SsPP2Cs were evolutionarily closer to that of sorghum, and the number of SsPP2Cs is” specify which analysis showed that.

Here in this study different techniques revealed functions of different genes which will be supportive for future studies, But is these analysis are sufficient to prove this? Without experimental data the evidence are weak.

Line 87-94 add economic and industrial significance of sugarcane.

All genes names must be italic.

Line 97 revise the sentence to remove grammatical error.

 Line 322 which studies? The studies should be cited

Phylogenetic structure analysis and gene structure analysis should be cited with recent studies. The following studies could be helpful https://doi.org/10.3390/ijms22179175, https://doi.org/10.1016/j.plaphy.2021.01.042

Add research gap and study limitations in conclusion.

Minor changes are required in some sentences to remove grammatical errors and typos

Author Response

Reviewer 1: This article revealed Genome-wide identification of the PP2C gene family and analyses with their expression profiling in response to cold stress in wild sugarcane. Before recommending this article for publication, there are some shortcomings for that should be resolve.

Abstract is well written however information of specific methods are not well presented.

Response: Thank you very much for this recommendation. We have added the other specific methods in our study.

Line 20 “SsPP2Cs were evolutionarily closer to that of sorghum, and the number of SsPP2Cs is” specify which analysis showed that.

Response: Thank you very much for this recommendation. We have revised the this sentence in the revised version as follows: Phylogenetic analysis suggested that SsPP2Cs are evolutionarily closer to that of sorghum, and the number of SsPP2Cs is the highest.

Here in this study different techniques revealed functions of different genes which will be supportive for future studies, But is these analysis are sufficient to prove this? Without experimental data the evidence are weak.

Response: Thanks for your suggestions. We have added the QPCR experiments to prove the transcriptomics data.

Line 87-94 add economic and industrial significance of sugarcane.

Response: Thanks for your suggestion. We have rewritten this economic and industrial significance of sugarcane, please refer to the introduction section of the revised MS.

All genes names must be italic. Line 97 revise the sentence to remove grammatical error.

Response: Thanks for your suggestion. We have corrected these errors. Moreover, the English writing of our MS has undergone English language editing by MDPI with the ID of 6666.

 Line 322 which studies? The studies should be cited

Response: Thanks for your suggestions. We have added these references.

Phylogenetic structure analysis and gene structure analysis should be cited with recent studies. The following studies could be helpful https://doi.org/10.3390/ijms22179175, https://doi.org/10.1016/j.plaphy.2021.01.042

Response: Thanks for your suggestions. The description on Phylogenetic structure analysis and gene structure analysis have been changed according to your suggestion.

Add research gap and study limitations in conclusion.

Response: Thanks for your suggestion and comments. The research gap and study limitations have been added in the revised conclusion.

Reviewer 2 Report

Huang et al. reported identification and expression profiles of the PP2C genes from sugarcane. However, most of researches were performed in silico, which provides only possibilities for all of issues. For example, heat map of low temperature-induced gene expression must be confirmed by the quantitative real time PCR. Similarly, confirmation of participation of Ca2+ and ABA in gene expression should be done by the quantitative real time PCR. Therefore, the manuscript is immature and the authors should consider additional experiments to improve the manuscript logically with results from wet experiments.

The authors predicted subcellular localization of PP2C members in silico and concluded the presence of chloroplast-, cytoplasm- and nuclear-localizing proteins. In addition, subcellular localization was shown in Fig. 10. However, these experiments showed only cytoplasmic and possible plasma membrane localization. It is problem that chloroplast- and nucleus-localization were not visualized. Moreover, cytoplasm of rice protoplast seems to close to plasma membrane, by which causes difficulty for judgement of cytoplasmic and plasma membrane localization in GFP reporter assays. Thus, if the authors hope to show the plasma membrane localization of SsPP2C31, the other evidence is required for making a conclusion.

Accordingly, conclusions represented in the manuscript is too speculative to understand gene expression profiles and subcellular localization. The authors must carefully redesign the experiments other than in silico analysis.   

Author Response

Reviewer 2: Huang et al. reported identification and expression profiles of the PP2C genes from sugarcane. However, most of researches were performed in silico, which provides only possibilities for all of issues. For example, heat map of low temperature-induced gene expression must be confirmed by the quantitative real time PCR. Similarly, confirmation of participation of Ca2+ and ABA in gene expression should be done by the quantitative real time PCR. Therefore, the manuscript is immature and the authors should consider additional experiments to improve the manuscript logically with results from wet experiments.

Response: Thanks for your suggestion and comments. The transcriptomics data of cold stress responded SsPP2C genes have been confirmed by QPCR in our revised manuscript, especially for these genes involved in the Ca2+ and ABA signal pathway, such as RCAR3, DIP2, PYL10, PYL5 and PYL7.

The authors predicted subcellular localization of PP2C members in silico and concluded the presence of chloroplast-, cytoplasm- and nuclear-localizing proteins. In addition, subcellular localization was shown in Fig. 10. However, these experiments showed only cytoplasmic and possible plasma membrane localization. It is problem that chloroplast- and nucleus-localization were not visualized. Moreover, cytoplasm of rice protoplast seems to close to plasma membrane, by which causes difficulty for judgement of cytoplasmic and plasma membrane localization in GFP reporter assays. Thus, if the authors hope to show the plasma membrane localization of SsPP2C31, the other evidence is required for making a conclusion. Accordingly, conclusions represented in the manuscript is too speculative to understand gene expression profiles and subcellular localization. The authors must carefully redesign the experiments other than in silico analysis.   

Response: Thanks for your suggestion and comments. We have added the QPCR results to confirm our transcriptomics data. Furthermore, as for the subcellular localization of PP2C members, especially for SsPP2C31, we have changed the description to: “whereas fluorescent signals of SsPP2C31 were found at the margin of the protoplast and the fluorescence signal intensity was significantly higher than that of other fusion proteins, suggesting that the SsPP2C31 protein might be cytoplasmic and plasma membrane localization protein.”

Reviewer 3 Report

the manuscript is very interesting and of good quality. But there are some results not very well represented and not adequqtely explained in the discussion . So the paper needs a minor revision 

Editing of english language is required

Author Response

Reviewer 3: the manuscript is very interesting and of good quality. But there are some results not very well represented and not adequqtely explained in the discussion . So the paper needs a minor revision. Comments on the Quality of English Language Editing of english language is required

Response: Thank you very much for your suggestion and comments. In our revised manuscript, we have revised the sections on Abstracts, Results, Discussions and Methods, as well as English writing of our MS has undergone English language editing by MDPI with the ID of 6666.

Round 2

Reviewer 2 Report

The authors performed quantitative real time PCR to confirm the results from the transcriptome analysis. However, there was no confirmation for the differences in time course of expression of PP2C genes during cold stress exposure. Since it is very important rather than correlation of fold induction as shown in Figure S4, experiments for comparison of temporal expression patterns under cold stress conditions should be performed to make a complete confirmation of the transcriptome data. Minor point: QPCR is not general expression. It should be q-PCR, q-PT-PCR, or quantitative real time PCR.

Author Response

Responded to reviewer: Thanks for your suggestions. qRT-PCR was performed to verify the reliability of the transcriptome data in time course of expression of PP2C genes during cold stress exposure. Besides, the QPCR has been changed to qRT-PCR.

Round 3

Reviewer 2 Report

The manuscript was significantly improved by addition of qRT-PCR data, which resolved problem I previously commented.